# MOSEV: a global burn severity database from MODIS (2000-2020)

Esteban Alonso-González[1], Víctor Fernández-García[2]

[1] Instituto Pirenaico de Ecología, Spanish Research Council (IPE-CSIC), Zaragoza, 50059, Spain
[2] Ecology, Department of Biodiversity and Environmental Management, University of León, León, 20971, Spain

*Correspondence to*: Víctor Fernández-García (vferg@unileon.es)

**Abstract.** To advance in the fire discipline as well as in the study of $CO_2$ emissions it is of great interest to develop a global database with estimators of the degree of biomass consumed by fire, which is defined as burn severity. In this work we present the first global burn severity database (MOSEV database), which is based on Moderate Resolution Imaging Spectroradiometer (MODIS) surface reflectance and burned area (BA) products since November 2000 to near real time. To

build the database we combined Terra MOD09A1 and Aqua MYD09A1 surface reflectance products to obtain dense time series of the Normalized Burn Ratio (NBR) spectral index, and we used the MCD64A1 product to identify BA and the date of burning. Then, we calculated for each burned pixel the difference of the NBR (dNBR), and its relativized version (RdNBR), as well as the post-burn NBR which are the most commonly used burn severity spectral indices. The database also includes the pre-burn NBR used for calculations, the date of the pre- and post-burn NBR and the date of burning. Moreover,

in this work we have compared the burn severity metrics included in MOSEV (dNBR, RdNBR and post-burn NBR) with the same ones obtained from Landsat-8 scenes, which have an original resolution of 30 m. We calculated the Pearson´s correlation coefficients and the significance of the relationships using 13 pairs of Landsat scenes randomly distributed across the globe, with a total BA of 6,904 $km^2$ (n = 32,163). Results showed that MOSEV and Landsat-8 burn severity indices are highly correlated, particularly the post-burn NBR (R= 0.88; P < 0.001). dNBR (R= 0.74; P < 0.001) showed stronger

relationships than RdNBR (R= 0.42; P < 0.001). Differences between MOSEV and Landsat-8 indices are attributable to variability in reflectance values and to the different temporal resolution of both satellites (MODIS: 1-2 days, Landsat-: 16 days). The database is structured according to the MODIS tiling system and is freely downloadable in https://doi.org/10.5281/zenodo.4265209 (Alonso-González and Fernández-García, 2020).

## 1 Introduction

More than half of the land surface on Earth can be affected by fire, an area about the size of the European Union being burned annually (Keeley et al., 2011; Moritz et al., 2012; Andela et al., 2019). Thus, fire is a phenomenon of great interest because its relevance worldwide but also because of expected changes in fire regimes as consequence of global warming and land use change (Moreira et al., 2020). Among these changes, previous work has reported that fire weather seasons have recently increased (18.7% from 1979 to 2013) (Jolly et al., 2015) whereas burned area (BA) has decreased globally (24,3%

from 1996 to 2015) (Andela et al., 2017; Forkel et al., 2019) mainly because the agricultural expansion in fire-dependent savannas (Andela et al., 2017).

The availability of satellite imagery with moderate spatial resolution (250-500 m) and daily or near-daily temporal resolution has enabled the production of several global BA products. Among the most accepted are those based on the Moderate Resolution Imaging Spectrometer (MODIS) (Chuvieco et al., 2016), which retrieves information of the entire Earth in 36

spectral bands every 1 to 2 days. The MODIS MCD64A1 C6 product (Giglio et al., 2018) is the standard NASA BA product, and probably the most used by the scientific community (Boschetti et al., 2019; Humber et al., 2019). MCD64A1 BA product is calculated with surface reflectance time series and fire active masks (Giglio et al., 2018), and was recently validated with Landsat imagery across the globe (stage-3 validation), reaching coefficients of determination above 0.70 despite an underestimation of small fires as consequence of its moderate spatial resolution (~500 m) (Boschetti et al., 2019).

Global BA products are essential to know the patterns of fire occurrence, fire extent, fire propagation (Rodrigues and Febrer, 2018) and fire frequency (Andela et al., 2019). Thus, BA products may be useful to provide an estimation of the global carbon emissions from biomass consumption (Veraverbeke et al., 2015; van der Werf et al., 2017). However, to go one step further in determining fire impacts on ecosystems as well as global carbon emissions it is necessary to characterize burned areas according to the degree of biomass consumption (Keeley, 2009; van der Werf et al., 2017).

The term used to define the degree of biomass consumption and the overall impact caused by fire on ecosystems is fire severity (preferred for field measurements) or burn severity (preferred for remote sensing measurements) (Keeley, 2009). Traditionally, burn severity has been quantified from Landsat sensors through different methods, including those based on radiative transfer models (Chuvieco et al., 2006, De Santis et al., 2009), spectral unmixing (Fernández-Manso et al., 2009, Quintano et al., 2017), or spectral indices (Chu and Guo, 2014, Fernández-García et al., 2018a). Among them, the standard

method to quantify burn severity is through the delta Normalized Burn Ratio (dNBR) (Key and Benson, 2006) spectral index, and its relativized version (RdNBR) (Miller and Thode, 2007), which is less dependent on the pre-fire vegetation, and potentially more suitable than dNBR for comparisons among zones with different environmental conditions (Miller and Thode, 2007; Rahman et al., 2018). Both spectral indices are based on the change caused by fire in near infrared (NIR) and shortwave infrared (SWIR) reflectance, which are highly sensitive to canopy density and moisture content respectively

(Chuvieco, 2010). dNBR and RdNBR indices calculated from Landsat have shown a high capacity ($R^2$ about 0.75) to correlate field measurements of biomass consumption and plant mortality in mediterranean (Fernández-García et al.., 2018a), temperate (Parks et al., 2014), boreal (Soverel et al., 2011) and tropical ecosystems (Rozario et al., 2018). Despite the possibility of calculating burn severity indices with satellites allowing planetary coverage such as MODIS (Veraverbeke et al., 2011; Rahman et al., 2018), there are not yet available products of burn severity at the global scale, which would be

useful to advance in fire and $CO_2$ sciences.

In this work we present a new burn severity database based on MODIS Terra and Aqua satellites. The presented database (MOdis burn SEVerity: MOSEV) provides monthly burn severity data (dNBR, RdNBR and post-burn NBR) with global

coverage since 2000 at 500 m spatial resolution. Additionally, this work describes the algorithm to develop the database and we compared the MOSEV burn severity data with their Landsat-8 equivalents.

## 2 MOSEV database

### 2.1 Input data

The MOSEV database was built using the following remote sensing data available since November 2000 as input (Fig. 1):

- All scenes of MODIS Terra MOD09A1 and Aqua MYD09A1 version 6: Terra MOD09A1 and Aqua MYD09A1 scenes are 8-day composites with 7 surface reflectance bands and quality information at spatial resolution of 500 m and global coverage. The reflectance value of each pixel is the best possible observation in the 8-day period, selected according to quality criteria including cloud cover, cloud shadow, solar zenith and aerosol loading.

- All scenes of MCD64A1 version 6 product: MCD64A1 is a monthly 500 m-pixel product that contains daily global information on burn date, uncertainty in burn date, quality assurance indicators and first and last day of the year of reliable change detection.

MOD09A1, MYD09A1 and MCD64A1 data was downloaded from the Land Processes Distributed Active Archive Center - (LP-DAAC): https://lpdaac.usgs.gov/ (last access: 1 November 2020).

### 2.2 Pre-processing

Terra MOD09A1 and Aqua MYD09A1 scenes were masked to remove water bodies, glaciers, clouds and snow. Masks were obtained directly from the MOD09A1 and MYD09A1 quality bands (surface reflectance 500 m band quality control flags). Likewise, MOD09A1 and MYD09A1 scenes not registering land surface were removed for subsequent analysis.

### 2.3 Algorithm overview

The method to obtain burn severity indices was structured in two steps (Fig. 1): (i) calculation of dense time series of the Normalized Burn Ratio (NBR) from merged Terra MOD09A1 and Aqua MYD09A1 scenes; (ii) selection of the pre- and post-fire NBR for each burned pixel and calculation of differenced burn severity indices (dNBR and RdNBR).

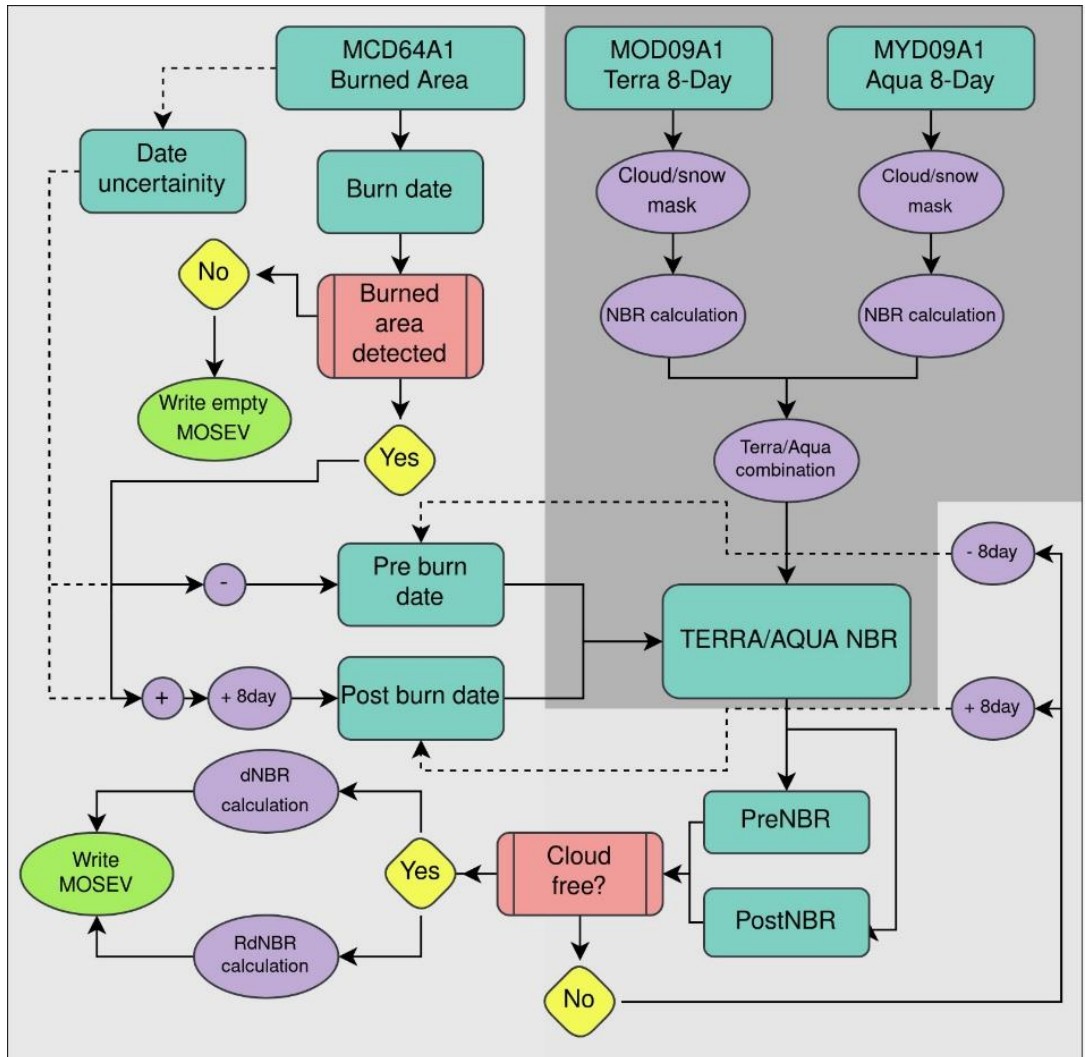

**Figure 1. Methodology flowchart used for building the MOSEV database (2000-present). MOD09A1 and MYD09A1 are 8-day reflectance products at 500 m from MODIS Terra and Aqua respectively. MCD64A1 is the monthly MODIS burned area product at 500 m spatial resolution. NBR, dNBR and RdNBR are burn severity spectral indices (Normalized Burn Ratio, difference of the NBR and Relativized dNBR respectively).**

**2.3.1 Calculation of dense time series of the Normalized Burn Ratio (NBR)**

The Normalized Burn Ratio (NBR) spectral index was calculated for each Terra MOD09A1 and Aqua MYD09A1 scenes according to the formula proposed by López-García and Caselles (1991) (Eq. 1). Terra NBR gaps (masked areas) were re-filled with the synchronous Aqua NBR values, when available. The combination of Terra and Aqua imagery is useful to reduce cloud contamination and therefore increase the data availability and decrease uncertainty (Yu et al., 2015;

Muhammad and Thapa, 2020). Thus, we obtained Terra/Aqua NBR composites with global coverage and a temporal resolution of 8 days since November 2000.

Eq.1. $NBR_{MODIS} = \frac{(\rho 2 - \rho 7)}{(\rho 2 + \rho 7)} \times 1000$

where ρ2 and ρ7 are the land surface reflectance values of bands 2 (841-876 nm) and 7 (2105-2155 nm) from Terra MOD09A1 and Aqua MYD09A1 scenes.

### 2.3.2 Selection of the pre- and post-burn NBR and calculation of burn severity indices

BA locations and dates were obtained from the MCD64A1 product. With the burn date and uncertainty in days, and considering the 8-day nature of our Terra-Aqua NBR, we have selected the immediate pre-burn (Eq. 2) and post-burn (Eq. 3) Terra/Aqua NBR dates for each MCD64A1 burned pixel.

Eq. 2. $preNBR\ date < MCD64A1\ burn\ day - MCD64A1\ uncertainty\ in\ days$

Eq. 3. $postNBR\ date > MCD64A1\ burn\ day + MCD64A1\ uncertainty\ in\ days + 8\ days$

When NBR values for the immediate pre-burn NBR date were not available (see 2.2 section) the previous NBR image was selected. On the contrary, when NBR values for the immediate post-burn date were not available the next NBR image was selected. These processes were repeated until pre- and post-burn NBR values were detected for each burned pixel in a cell by cell basis.

We have obtained the pre-burn NBR value and the post-burn NBR value from the pre- and post-fire Terra/Aqua NBR dates, which were used to calculate the dNBR and RdNBR value for each burned pixel of the MCD64A1 product. Both, dNBR and RdNBR are bi-temporal spectral indices that account for the change caused by fire in NIR and SWIR reflectance.

dNBR is the reference burn severity spectral index used by the European Forest Fire Information System (https://effis.jrc.ec.europa.eu/about-effis/) and by the Monitoring Trends in Burn Severity program of the US 115 (https://www.mtbs.gov/), and was calculated according to Key and Benson (2006) (Eq. 4), dNBR values increasing with burn severity.

Eq. 4. $dNBR = preNBR - postNBR$

Likewise, RdNBR is also an outspread burn severity spectral index, used by the Monitoring Trends in Burn Severity program of the USA (https://www.mtbs.gov/). RdNBR was calculated according to Miller and Thode (2007) (Eq. 5), higher 120 RdNBR values indicating higher burn severity.

Eq. 5. $RdNBR = \frac{dNBR}{\sqrt{\frac{|preNBR|}{1000}}}$

## 2.4 Implementation

Burn date from MCD64A1, pre-burn NBR date, post-burn NBR date, pre-burn NBR, post-burn NBR, dNBR and RdNBR were written in monthly scenes since November 2000 at spatial resolution of 500 m (MOSEV database). All operations to calculate and write the database were carried in R programming language using the rspatial/luna (Ghosh et al., 2020) and terra (Hijmans et al., 2020) libraries and Bash Unix shell command language. All the calculations and data manipulation were performed in the supercomputing facilities of the Spanish Research Council (CSIC).

## 2.5 Comparison with Landsat burn severity indices

In order to evaluate the MOSEV database, we have compared MOSEV burn severity indices (dNBR, RdNBR and post-burn NBR) with the same indices manually obtained from higher spatial resolution imagery. To perform the comparison, we have selected Landsat scenes, which have 30 m spatial resolution and have been the most used imagery for burn severity assessments (Key and Benson, 2006; Fernández-García et al., 2018a). We selected 13 regions of 185 km x 180 km (Landsat-8 tile dimension) with a large extent of BA, and randomly distributed across the globe (See Table A1 in the Appendix A). Pre- and post-burn consecutive scenes (16 days span) with low cloud cover ($< 25\%$) of the Landsat-8 Collection 1 Level-2 product were selected for each region and downloaded from the USGS Earth Explorer (https://earthexplorer.usgs.gov/ last access: 1 November 2020). Landsat-8 Collection 1 Level-2 scenes are composed of 7 land surface reflectance bands at a spatial resolution of 30 m, and a quality band which was used to mask cloud covered areas. Bands 5 (850-880 nm) and 7 (2110-2290 nm), which are comparable to MODIS bands 2 (841-876 nm) and 7 (2105-2155 nm), were aggregated and resampled averaging the Landsat values to the MODIS grid, in order to match the spatial resolution of the MOSEV products (500 m). Landsat-8 resampled bands were used to calculate the pre-burn NBR and the post-burn NBR (Eq. 6) as well as the dNBR (Eq. 4) and RdNBR (Eq. 5) spectral indices.

Eq.6. $\quad NBR_{OLI} = \frac{(\rho 5 - \rho 7)}{(\rho 5 + \rho 7)} \times 1000$

where ρ5 and ρ7 are the land surface reflectance values of Landsat 8 OLI/TIRS bands 5 (850-880 nm) and 7 (2110-2290 nm) resampled to the spatial resolution of MOSEV products.

To assess the relationships between the burn severity indices included in the MOSEV database with the same ones derived from Landsat-8, we sampled all available burned pixels (n = 32,163) of the 13 study regions from both, MOSEV and Landsat-8 dNBR, RdNBR and post-burn NBR layers (Table A1). Then, we performed scatterplots and we calculated the Pearson´s correlation coefficients (R) and the significance of the correlations (P).

## 3 Data description

The MOSEV database ([https://doi.org/10.5281/zenodo.4265209](https://doi.org/10.5281/zenodo.4265209) Alonso-González and Fernández-García, 2020) is composed of monthly scenes since November 2000 with a spatial resolution of 500 m. The scenes are organized following the tiling system of MODIS (sinusoidal tile grid). In total, the database is structured in 246 non-overlapping tiles that cover an area of 10 degrees by 10 degrees in the equator (Fig. 2). The name of each MOSEV scene encodes the year, Julian day indicating the month, and MODIS tile. For instance, the MOSEV. A2019305.h31v11 scene corresponds to the year 2019, month of

November (month ended in the Julian day 305) and h31v11 MODIS tile.

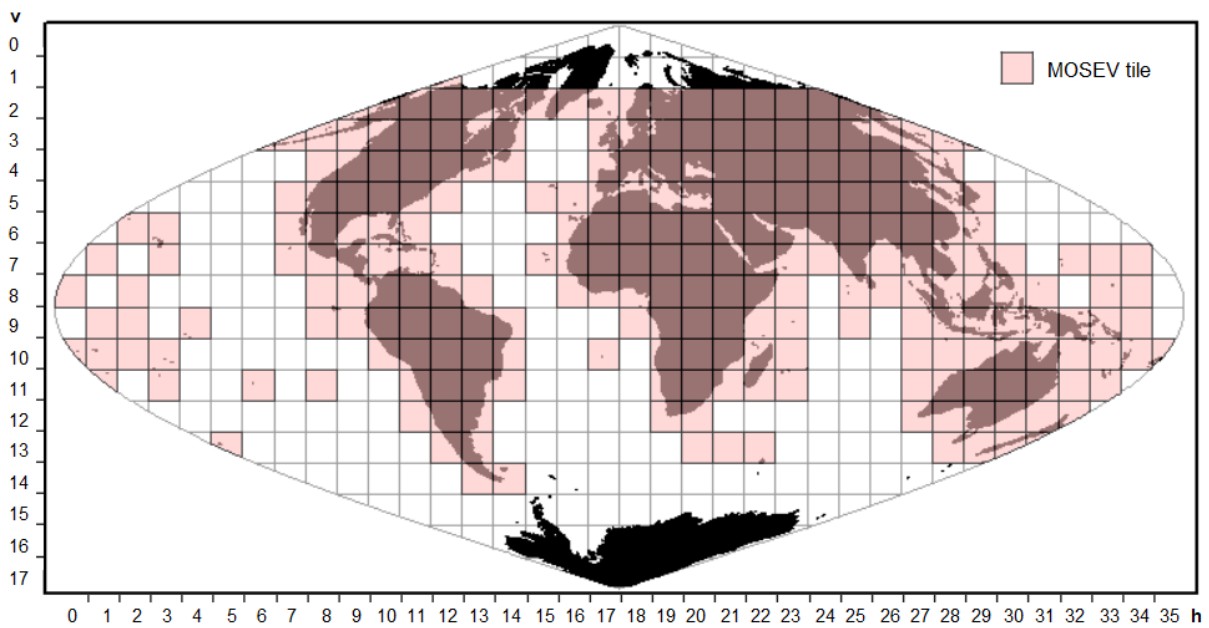

**Figure 2. MODIS sinusoidal tiling system and available MOSEV tiles.**

Each MOSEV scene is composed of 7 layers (Table 1; Fig. 3): dNBR, RdNBR, pre-burn NBR, post-burn NBR, pre-burn

selected date, post-burn selected date and the burn date from MCD64A1. In all layers we assigned the values of -32767 to unburned land, 32767 to water bodies, and a value of -18000 was assigned to those areas where was not possible to fill with a severity value or the severity value was out of the allowed range.

- dNBR: the valid range in the MOSEV database corresponds to their mathematical feasible range (-2000 to 2000) (see Eq. 4), although values above 1200 are not usual (Key and Benson, 2006).

- RdNBR: the valid range in the MOSEV database was bounded from -4000 to 4000 since values outside these limits are feasible (see Eq. 5) but anomalous (Miller and Thode, 2007; Miller et al., 2009).

- Pre- and post-burn NBR: both spectral indices are the Terra/Aqua composites used in the calculation of dNBR and RdNBR. Likewise, the post-burn NBR is the most common mono-temporal burn severity spectral index, decreasing its value as burn severity increases. The pre- and post-burn NBR valid range in the MOSEV database corresponds to their mathematical

feasible range (-1000 to 1000) (see Eq. 1).

- Pre- and post-burn selected dates: they are estimators of the pre- and post-burn NBR dates and represent the number of iterations necessary to find available pre- and post-burn NBR values. Specifically, a value of 0 in the selected date indicates that the NBR date is the immediate NBR according to the equations 2 (pre-burn) and 3 (post-burn). A value of 1 indicates that the immediate NBR value was not available, and the previous (in the case of the pre-fire) or the next (in the case of post-

fire) NBR value was used instead.

- Burn date from MCD64A1: is the date of burning in Julian days registered in the MCD64A1 BA product. It was used as basis to identify the pre- and post-burn selected dates and pre- and post-burn NBR values.

In order to reduce the overall size of the database, the MOSEV scenes where no fires were detected are composed by a single empty layer entitled "burndate_from_MCD64A1". We did it in this way, in order to maintain the same number of MOSEV

files per tile, even in the unburned scenarios, similarly to the original MCD64A1 product. Each MOSEV file is a multi or single band GeoTIFF in 16 bit integer, compressed using the lossless compression algorithm Lempel–Ziv–Welch (LZW). The MOSEV scenes are distributed as a zipped file, constituted by all the scenes of each tile. The complete dataset can be freely downloaded at https://doi.org/10.5281/zenodo.4265209 (Alonso-González and Fernández-García, 2020).

**Table 1. Layers of the MOSEV product.**

| Layer | Units | Type | Valid range | Unburned land | Water | No data |
|-------|-------|------|-------------|---------------|-------|---------|
| dNBR | unitless | 16 bit | -2000 to 2000 | -32767 | 32767 | -18000 |
| RdNBR | unitless | 16 bit | -4000 to 4000 | -32767 | 32767 | -18000 |
| preNBR | unitless | 16 bit | -1000 to 1000 | -32767 | 32767 | -18000 |
| postNBR | unitless | 16 bit | -1000 to 1000 | -32767 | 32767 | -18000 |
| preburn_selected_date | cycles | 16 bit | $\geq 0$ | -32767 | 32767 | -18000 |
| postburn_selected_date | cycles | 16 bit | $\geq 0$ | -32767 | 32767 | -18000 |
| burndate_from_MCD64A1 | days | 16 bit | 1 to 366 | -32767 | 32767 | -18000 |


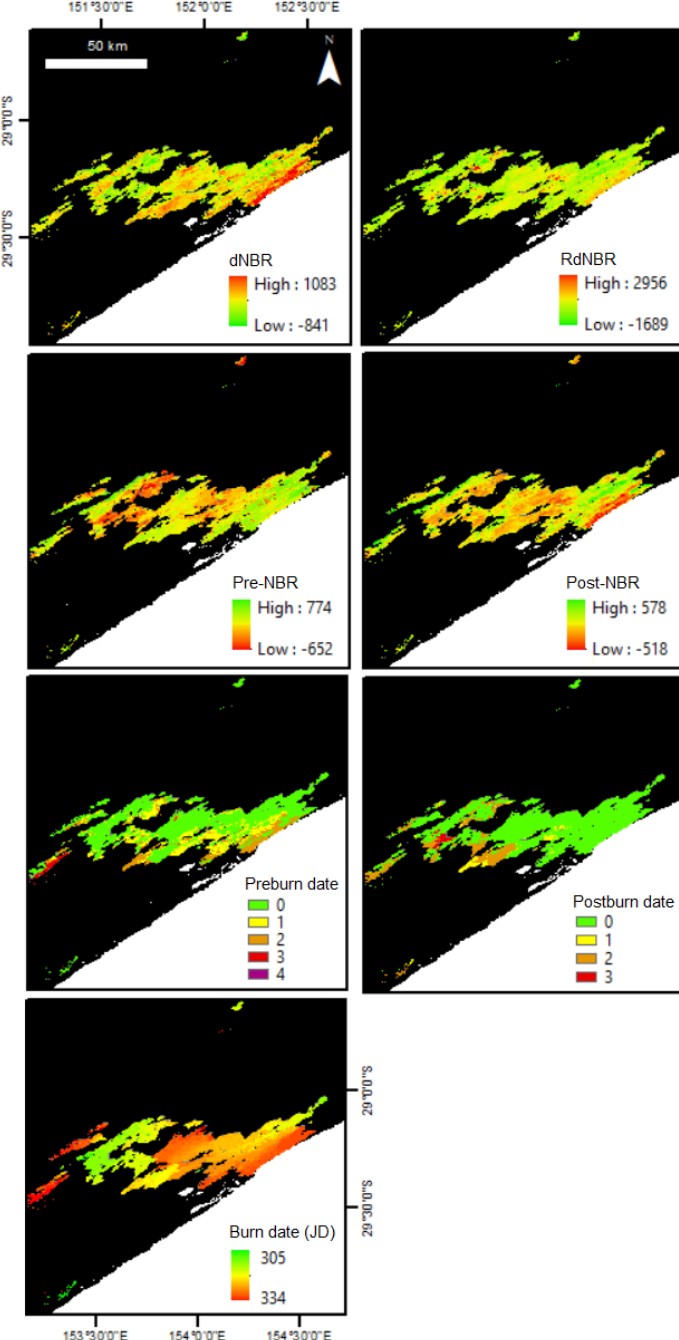

**Figure 3. Example of the layers included in a MOSEV scene (A2019305.h31v11) representing some of the 2019 wildfires in Australia (New South Wales). Spectral indices dNBR, RdNBR, Pre-NBR and Post-NBR are unitless. The pre-burn and post-burn dates indicate the number of cycles or iterations necessary to find available NBR values (each cycle added to 0 corresponds with a difference of 8 days). The burn date is expressed in Julian days. White areas are water bodies (value of 32767) and black areas are unburned land (value of -32767).**


# 4 Results and discussion

In this work we have developed the MOSEV product, which is a global burn severity database based on MODIS Terra and Aqua surface reflectance and MODIS BA products. The database includes dNBR, RdNBR and NBR burn severity indices at 500 m pixel size, which are usually calculated at local scale using higher resolution imagery, traditionally with Landsat scenes (Chuvieco, 2010; Key and Benson, 2006; Miller and Thode, 2007; Fernández-García et al., 2018a) and more recently with Sentinel-2 scenes (Fernández-Manso et al., 2016).

## 4.1 Comparison with Landsat burn severity indices

The probability density functions of MOSEV and Landsat burn severity indices (dNBR, RdNBR and post-burn NBR), as well as their relationships are shown in Fig. 4. In general, results showed a high similarity between MOSEV and Landsat probability density functions, with values more concentrated to the mean for Landsat indices, and a slight negative bias for the three spectral indices, as MOSEV mean values tended to be lower than Landsat data (Table 2). MOSEV and Landsat burn severity indices were highly correlated (P < 0.001) for the three burn severity indices. Specifically, the post-burn NBR showed higher correlation coefficient (R = 0.88) than dNBR (R= 0.74) and RdNBR (R = 0.42).

Previous research has found discrepancies in surface reflectance when comparing both, MODIS and Landsat satellites (Feng et al., 2013; Veraverbeke et al., 2011; Ke et al., 2015; Potapov et al., 2020). These differences could be explained by several reasons: (i) the higher temporal resolution of MODIS imagery used to build the database enables to use pre-fire information very close to the burning event. This has a significant influence in the pre-burn NBR values, which typically decrease as the fire season approaches and drought conditions are more severe (Wang et al., 2008). This fact explains the lower pre-burn NBR values and the higher proportion of negative values in the MOSEV product compared with Landsat (Figure A1, Table A2), which unavoidably lead to some differences in dNBR and RdNBR values and could contribute to a higher proportion of negative values in these indices in the MOSEV database (Table 2). Also the post-fire information in the MOSEV product is very close to the burning event, thus potentially allowing a better assessment of burn severity compared with the lower temporal resolution of the Landsat constellation. (ii) Potential errors in radiometric gains from Landsat imagery, which are used for rescaling digital numbers to radiance values (Chander et al., 2009). (iii) Saturation problems in bright surfaces have been detected by Feng et al. (2013) in Landsat imagery but not for MODIS. This effect may influence the quality of the pre-fire NIR and the post-fire SWIR reflectance, which have high values in severely burned areas (Key and Benson, 2006). (iv) Differences in imagery pre-processing may affect the final reflectance values. In this sense, Landsat imagery is resampled using a cubic convolution method (uses 16 nearest-neighbor data points) in the geometric correction stage (Landsat 8 Data Users Handbook Version 5.0, 2019), whereas MODIS reflectance products are resampled using bilinear interpolation (4 nearest-neighbor data points) (MODIS Science Data Support Team, 1997). The use of cubic convolution method smooths reflectance values more than bilinear interpolation, contributing to moderate extreme values. This assumption is supported by the probability density functions (Fig. 4), which show higher abundance of extreme values for the spectral indices when

calculated from MODIS instead of Landsat. Similarly, it was necessary to resample the Landsat products to the MODIS grid

to make it comparable considering the big resolution shift, introducing some obvious and unavoidable uncertainty.

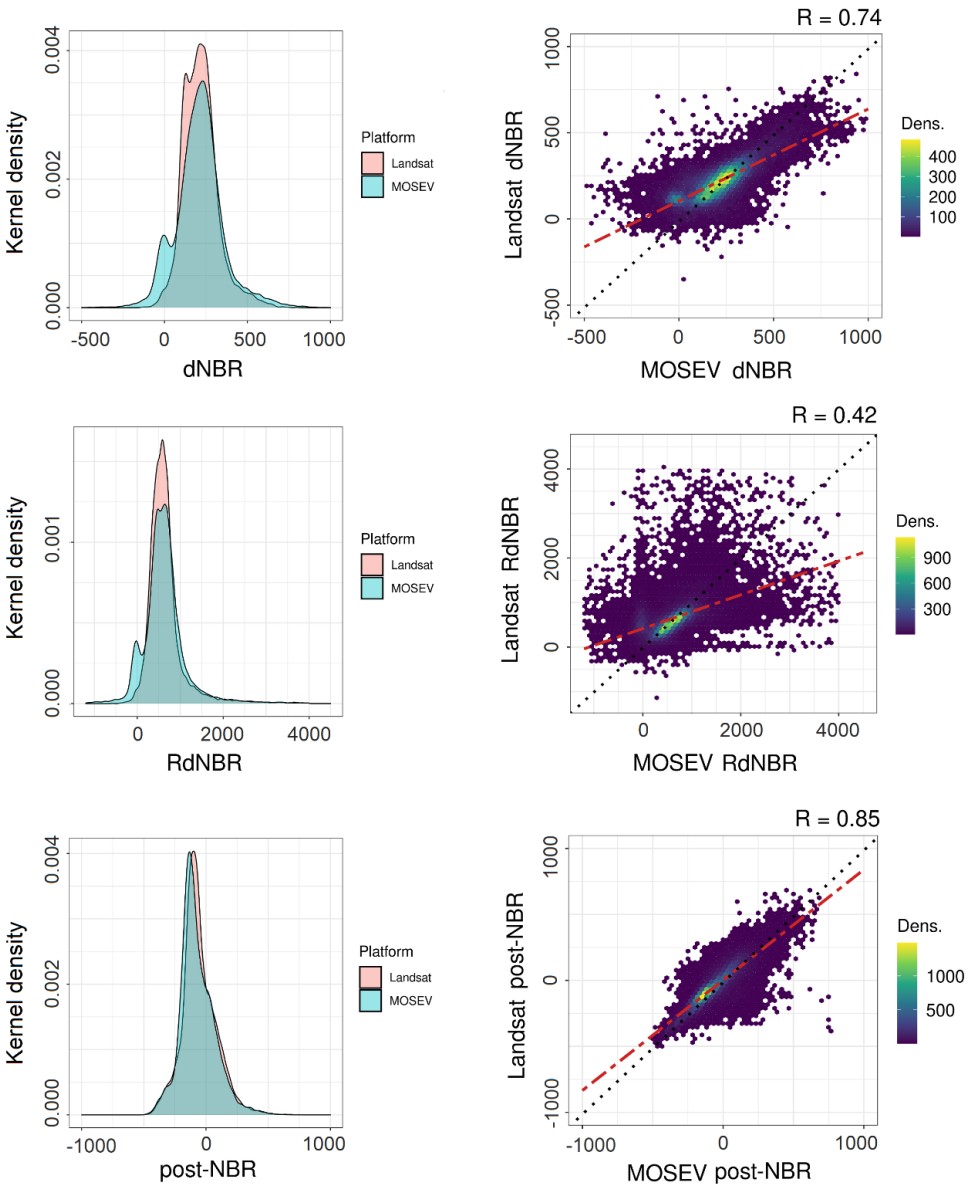

**Figure 4. Gaussian kernel densities (left) and scatterplots (right) showing the relationships between the burn severity data included in the MOSEV database (dNBR, RdNBR and post-burn NBR spectral indices) and the equivalent obtained from 13 Landsat-8 scenes randomly distributed across the globe (n = 32,163). R: Pearson´s correlation coefficient. Dens. = density of points. See table**

**A1 for further information.**

**Table 2. Mean values, and proportion of positive and negative values of the burn severity spectral indices included in the MOSEV database (dNBR, RdNBR and post-burn NBR spectral indices) and the equivalent obtained from 13 Landsat-8 scenes randomly distributed across the globe (n = 32,163).**


| Layer | MOSEV (mean) | Landsat (mean) | MOSEV (% positive) | Landsat (% positive) | MOSEV (% negative) | Landsat (% negative) |
|---|---|---|---|---|---|---|
| dNBR | 210.3 | 216.7 | 92.0 | 98.8 | 8.0 | 1.2 |
| RdNBR | 616.0 | 651.5 | 92.0 | 98.8 | 8.0 | 1.2 |
| postNBR | -68.4 | -54.7 | 27.4 | 29.4 | 72.6 | 70.6 |

Likewise, we detected variability in the correlations between MOSEV and Landsat among the three studied burn severity indices. The higher correspondence of the pre-burn and post-burn NBR spectral indices suggests that the mono-temporal approach contribute to achieve higher correlations (see post-NBR in Fig. 4 and pre-NBR in Fig. A1). On the contrary,
combining the information of two scenes (bi-temporal approach) the higher differences in satellite reflectance's augmented differences in bi-temporal spectral indices (dNBR and RdNBR). Focusing on dNBR and RdNBR, we found higher correlations between MOSEV and LANDSAT dNBR than RdNBR, which can be direct consequence of the RdNBR algorithm (see Eq. 5), as low pre-burn NBR absolute values may cause really high and even anomalous RdNBR values, generating heteroscedasticity (see Fig. 4). However, in general both burn severity products showed good levels of agreement,
considering the unavoidable uncertainties associated with the very different nature of Landsat and MODIS. Thus, the comparison of MOSEV and Landsat burn severity prove the consistency of the developed algorithm, being the dNBR relationships similar (Veraverbeke et al., 2011) or even better (Rahman et al., 2018) than those found in previous studies that compared retrieved burn severity information from both products.

### 4.2 Indices interpretation and validity

NBR, dNBR and RdNBR spectral indices were developed to provide optimum measurements of fire effects and biomass consumption from multispectral imagery (Key and Benson, 2006; Miller and Thode, 2007). The three indices are based on the change caused by burning on NIR and SWIR reflectance. NIR reflectance is highly sensitive to canopy density, being lower in burned than in vegetated areas. On the contrary, SWIR reflectance is sensitive to moisture content and ash, and is significantly higher in burned areas than in vegetated zones. The NBR index (López-García and Caselles, 1991) uses this
information to estimate burn severity from a mono-temporal approach. In general, high positive NBR values indicate vegetated areas, whereas bare soil and burned areas present low and usually negative NBR values, the more negative the

more severely burned. The validation of burn severity indices has been widely addressed using Landsat imagery and field measurements at similar spatial resolution (~ 30 m), because the high difficulty of taking accurate field measurements at coarser spatial scales. The most popular field measurement to assess the performance of burn severity spectral indices are

those indices based on the Composite Burn Index (CBI) (Key and Benson, 2006), which combine information of fuel consumption and related changes caused by fire, including plant mortality, char height and soil colour. In general, the literature shows high variability in the goodness-of-fit between NBR and CBI-type indices. For instance, Picotte and Robertson (2011) found $R^2$ values ranging from 0.61 to 0.86 in different ecosystems across North America the following months after fire; De Santis and Chuvieco reported $R^2$ values between 0.32-0.66, largely varying depending on the CBI-type

index used; and Fernández-García et al. (2018) obtained $R^2$ values between 0.69 and 0.88 for different forest ecosystems in the Iberian Peninsula. Although NBR has proven capacity to indicate burn severity, it is usually overcome by differenced indices such as the dNBR and RdNBR, which account for the pre-burn conditions (Key and Benson, 2006; Fernández-García et al., 2018).

The dNBR and RdNBR indices provide quantitative measurement of the environmental change between the pre- and post-

burn situation (i.e. biomass consumption and related changes) (Key and Benson, 2006). The dNBR is a measurement of severity understood as absolute change, whereas the RdNBR was designed to relativize the dNBR to the pre-fire situation, thus, the total combustion of areas with different amount of vegetation would lead to similar RdNBR values, but dNBR would be higher in the most vegetated area (Miller and Thode, 2007). The performance of both, dNBR and RdNBR indices obtained from Landsat has been validated in numerous studies across the globe. For instance, Zhu et al. (2006) reported

mean $R^2$ values of 0.67 (dNBR) and 0.60 (RdNBR) when correlating CBI in different ecosystems in North America; Parks et al. (2014) found mean $R^2$ values of 0.76 (dNBR) and 0.77 (RdNBR) in Western United States; Fernández-García et al (2018) used a CBI-type index to analyse the goodness-of-fit of both, dNBR ($R^2 = 0.81$) and RdNBR ($R^2 = 0.78$) indices in forest ecosystems in the Iberian Peninsula; Cai and Wang (2020) found a better performance of dNBR ($R^2 = 0.84$) than RdNBR ($R^2 = 0.79$) when correlating a descriptive burn severity index in south-east China; and Rozario et al. (2018) found that

dNBR ($R^2 = 0.56$) and RdNBR ($R^2 = 0.58$) indices were able to indicate and the percentage of scorched vegetation in tropical dry forests of Costa Rica.

The values of both, dNBR and RdNBR increase proportionally to burn severity, and in general, values below zero indicate unburned or recovered areas. dNBR and RdNBR continuous data can be used to differentiate burn severity categories. The number of classes and burn severity thresholds is entirely up to the user's objective, and empirical thresholding based on the

relationships with field data is useful to provide ecological meaning to the spectral index-based categories (Key and Benson, 2006; Fernández-García et al., 2018). An option when field data is not available is to define thresholds according to the literature (Rozario et al., 2018). In this sense, Key and Benson (2006) provided six dNBR thresholds to differentiate seven burn severity categories, whereas Miller and Thode (2007) provided three thresholds (41, 176 and 366 for dNBR; 69, 315, 640 for RdNBR) to differentiate unchanged, low, moderate and high severities in forest ecosystems of North America with

the post-fire image taken one year after fire. Preferred threshold values are higher when the post-fire image is closer to the

fire date (Key and Benson, 2006), as it is the case of MOSEV indices. Examples of thresholds in forest ecosystems with the post-fire images taken immediately after fire are those provided by Botella-Martínez and Fernández-Manso (2017), which differentiated unburned, low, moderate and high with three threshold values (160, 260, 481 for dNBR; 230, 475, 835 for RdNBR).

**4.3 Advancements and limitations**

The main asset of MOSEV database is that it is the first global burn severity database, which complement the existing global BA data such as the FireCCI50 (Chuvieco et al., 2018) or the MCD64A1 C6 product (Giglio et al., 2018). One of the most important strengths of MOSEV is consequence of MODIS revisit time (1 to 2 days), which is shorter than Landsat-8 (16 days) and Sentinels-2 (5 days). This high temporal resolution allowed us to obtain dense free-cloud NBR time series that can
be indispensable to calculate burn severity indices in regions of persistent cloud cover. In fact, Ju and Roy (2008) show that the probability of finding two consecutive Landsat scenes within a month is 0.63 globally, but near 0 in many regions such Russia and Canada, and many areas of Central Africa among others. Likewise, another improvement of MOSEV burn severity indices over indices calculated from other satellites such as Landsat or Sentinel is the higher temporal consistency of the data, as Terra and Aqua satellites use the same MODIS sensor since 2000.
The main limitation of MOSEV database is related to its spatial resolution of 500 m, which impedes to account for fine-grain spatial heterogeneity. However, this spatial resolution enables the study of burn severity at regional and planetary scale with low computational costs. Another fact to consider is the error in the classification of burned areas in the MCD64A1 BA product in which MOSEV is based. In this sense, Giglio et al. (2018) reported a global Commission Error (CE) of 24% and an Omission Error (OE) of 37%, whereas Boschetti et al. (2019) in a Stage-3 validation indicated a global CE of 40% and an
OE of 73%. The lowest errors were detected in regions where fires are larger, and fire scars persistent, such as in boreal forests.

**4.4 Potential applications**

Burn severity metrics from the MOSEV database can be useful to analyse temporal trends in burn severity, to study the global spatial patterns of burn severity, to identify areas where the post-fire recovery of soil and vegetation can be
endangered, and to enhance global models of carbon emissions among other applications. In addition, it will constitute a cost-effective way of monitoring the global burn severity in a close to real time way, as MOSEV could be upgraded with the same temporal frequency of the MCD64A1 product.

In relation to the temporal trends of burn severity, it is common in the fire ecology literature to assume increases in burn severity owing to climate change (e.g. García-Llamas et al., 2019; Moreira et al., 2020). However, there is little evidence of
that at the planetary scale since there was not global burn severity data. Previous studies in that line have analysed temporal trends in burn severity at the regional scale, mainly in the USA and Europe (Fried et al., 2004; Parks et al., 2016; Picotte et

al., 2016). With the MOSEV database it is possible to study global trends in burn severity and study relationships between burn severity and global change.

Spatial patterns of fire occurrence and burn severity have also captured the interest of several researchers (e.g. Duffy et al., 2007; Kennedy and Johnson, 2014; Stevens et al., 2017), but research at the global scale is limited to the study of BA (Andela et al., 2017). Thus, the MOSEV database opens the possibility of expanding the study of fire patterns to the planetary scale including the variable burn severity.

Burn severity is a variable of high interest to predict ecosystem responses (Keeley, 2009). Among the most relevant ecosystem responses for forest managers is soil erosion (De Luis et al., 2003) and vegetation recovery (Fernández-García et al., 2018b; 2019; 2020). Thus, MOSEV burn severity indices may serve as a tool for land managers to roughly identify target areas for post-fire forest management, as well as to study predictors of burn severity which could be useful for pre-fire management (García-Llamas et al., 2019).

Moreover, previous work has shown the importance of including burn severity metrics as predictors of $CO_2$ emissions caused by fires (e.g. Veraverbeke et al., 2015; van der Werf et al., 2017). The MOSEV database will be useful for the enhancement of global $CO_2$ emission models.

## 5 Data availability

The MOSEV database is freely downloadable in https://doi.org/10.5281/zenodo.4265209 (Alonso-González and Fernández-García, 2020).

## 6 Conclusions

We have introduced the newly developed MOSEV database, which is the first burn severity database with global coverage, available since November 2000. The algorithm used to build the database is based on MODIS Terra and Aqua surface reflectance imagery, as well as on the MCD64A1 BA product. MOSEV data includes seven layers at 500 m pixel size with the most commonly used burn severity spectral indices (dNBR, RdNBR and post-burn NBR), the pre-burn NBR, estimators of the date of the pre- and post-burn MODIS surface reflectance scenes used for calculations and the date of burning. The burn severity indices from MOSEV showed consistent relationships with Landsat-derived burn severity indices, which have been the most used for burn severity assessments. Thus, this database could be the base to accomplish future studies of burn severity at the global scale, in a computational cost-effective way, as well as research where burn severity could be a relevant factor such as in forest management and $CO_2$ emissions research.

**Appendix A**

Table A1. Scenes and number of pixels (n) used to compare MOSEV and Landsat-8 (L8)-derived burn severity indices (dNBR, RdNBR and post-burn NBR).

| Location | n | MOSEV scene | Pre-burn L8 scene | Post-burn L8 scene |
|---|---|---|---|---|
| Brazil | 450 | MOSEV.A2019213.h11v09 | LC082320652019081201T1-SC20200607111702 | LC082320652019082801T1-SC20200607111630 |
| Nepal | 1,082 | MOSEV.A2019121.h25v06 | LC081430402019050901T1-SC20200607111858 | LC081430402019052501T1-SC20200607111838 |
| USA | 133 | MOSEV.A2019274.h08v05 | LC080370362019100901T1-SC20200607111820 | LC080370362019102501T1-SC20200607111815 |
| Russia | 2,188 | MOSEV.A2019182.h24v02 | LC081170172019072201T1-SC20200607111752 | LC081170172019080701T1-SC20200607111749 |
| Senegal | 9,245 | MOSEV.A2019032.h16v07 | LC082040512019021301T1-SC20200607111948 | LC082040512019021301T1--SC20200607111808 |
| Kazakhstan | 2,091 | MOSEV.A019244.h22v03 | LC081510252019092201T1-SC20200607111850 | LC081510252019100801T1-SC20200607111857 |
| Zambia | 8,863 | MOSEV.A2019182.h20v10 | LC081730702019071401T1-SC20200607111924 | LC081730702019073001T1-SC20200607111814 |
| Bolivia | 83 | MOSEV.A2019182.h11v10 | LC080010712019070901T1-SC20200607111740 | LC080010712019072501T1-SC20200607111831 |
| Canada | 17 | MOSEV.A019182.h12v02 | LC080610152019062701T1-SC20200607111838 | LC080610152019071301T1-SC20200607111857 |
| South Africa | 371 | MOSEV.A2019001.h19v12 | LC081750842019010101T1-SC20200607111855 | LC081750842019011701T1-SC20200607111857 |
| Kazakhstan | 5,130 | MOSEV.A2019182.h23v04 | LC081520272019071101T1 -SC20200607111844 | LC081520272019072701T1-SC20200607111833 |
| Mozambique | 1,103 | MOSEV.A2019213.h21v10 | LC081670742019080501T1-SC20200607111843 | LC081670742019082101T1-SC20200607111845 |
| Russia | 1,407 | MOSEV.A2019091.h19v03 | LC081880222019040201T1-SC20200607111738 | LC081880222019041801T1-SC20200607111829 |
| Total | 32,163 | | | |

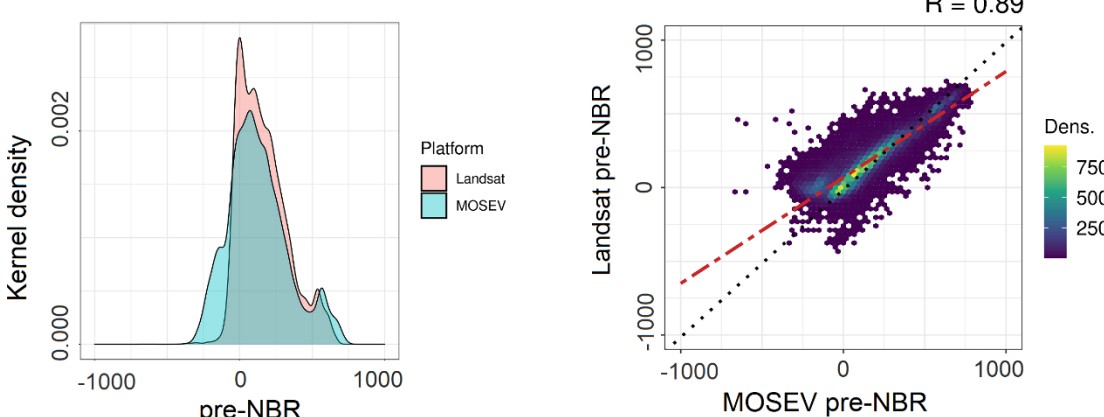

**Figure A1. Gaussian kernel densities (left) and scatterplots (right) showing the relationships between the pre-burn NBR spectral index included in the MOSEV database and the equivalent obtained from 13 Landsat-8 scenes randomly distributed across the globe (n = 32,163). R: Pearson´s correlation coefficient. Dens. = density of points.**

**Table A2. Mean values, and proportion of positive and negative values of the pre-burn NBR spectral index included in the**
**MOSEV database and the equivalent obtained from 13 Landsat-8 scenes randomly distributed across the globe (n = 32,163).**

| Layer | MOSEV (mean) | Landsat (mean) | MOSEV (% positive) | Landsat (% positive) | MOSEV (% negative) | Landsat (% negative) |
|---|---|---|---|---|---|---|
| preNBR | 128.5 | 161.0 | 72.2 | 82.7 | 27.8 | 17.3 |

**Author contribution**

EA-G and VF-G designed the database and the structure of the manuscript. EA-G developed the algorithms and the MOSEV
database. VF-G prepared the Landsat-8 data. EA-G did the statistical analysis of the data. EA-G and VF-G made the figures. VF-G wrote the first draft of the manuscript and EA-G contributed to write the manuscript.

**Competing interests**

The authors declare that they have no conflicts of interest.

**Financial support**

This project has not received funding.

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
