# Peer review of "MOSEV: a global burn severity database from MODIS (2000-2020)"

_Earth System Science Data, 2020_

## Referee Comment (RC1) · Anonymous Referee #1 · 7 Jan 2021

General comment

The manuscript presents a newly developed global database of MODIS-based burn severity indices for the 21st century. I believe that such readily available dataset is an important development which would enable researchers to perform a range of analyses focused on global fire dynamics and impacts. I can certainly see myself using the dataset in the future. The manuscript is generally well structured and written, however, I have a few concerns and comments which I would like the authors to address before publication.

My main critique of the manuscript is twofold. Firstly, the presented comparison of

the MOSEV vs Landsat burn severity estimates is very limited and offers little in terms of actual validation and uncertainty estimation. While I do realize that validating such a dataset is difficult to say the least, but with little extra work the authors could help the potential users of the MOSEV dataset to gauge magnitude of uncertainties and biases associated with the burn severity estimates. In particular, it would be useful to see distributions of the burn severity indices in addition to the presented comparison data. Secondly, the manuscript lacks discussion on what typical satellite-based pre- and post- burn NBR (and hence dNBR or RdNBR) values are and how they relate to biomass consumption/fire severity as measured by field studies or any other methods. Please see the specific comments for further details.

Specific comments

Lines 55 – 57: The sentence is the only place that I could see where a quantitative estimate of how well satellite-based burn severity estimates relate to field data. This is key information and the discussion should be expanded. It is currently not clear if these published dNBR and RdNBR validations against field estimates of biomass consumption and plant mortality apply to Landsat data? Where ther any MODIS data-based validations? Also please provide estimates for different environments if available. This information is critical in supporting the undertaken comparison of MOSEV dataset with Landsat estimates (which serves as the only validation at this point).

Line 85 (Figure 1) What does "Fires Detected" conditional block represent in the flowchart? Is this simply checking if MCD64A1 tile contain burned pixels of something else? If additional filtering to MCD64A1 data was applied, what were the implications for MOSEV coverage in comparison to MCD64A1 burned area?

Lines 100 – 105: From the description it is not clear why burned area date uncertainty is inflated by 8 days in Eq. 3, but the same is not done when determining preNBR date in Eq. 2. Could authors explain this.

Lines 163 – 170: The section presents mathematical ranges of the data, however it

would be also very interesting to see distributions of actual retrieved values, perhaps in the results section. In particular, what proportion of total dNBR and RdNBR estimates have negative values and what is the interpretation of this result. This is very interesting as MCD64A1 burned area algorithm uses difference in reflectance of bands p5 and p7 (p7 is also used by MOSEV) to determine burned area in the first place.

Line 178: "no fires were detected" is ambiguous as the term is usually associated with active fire satellite products. Was active fire detection data used in the study? If not, perhaps "no burned area pixels" would be more clear.

Lines 202 – 203: I'm not sure if the scatterplots (Fig. 4) make it possible to tell weather there's a positive bias in MOSEV estimates vs Landsat. To me it seems that at lower values MOSEV estimates are lower, and that this results in the slopes of linear fits seen in Fig. 4. For dNRB subplot in particular I see a dense cluster of negative MOSEV dNRB values which correspond to low (but positive) Landsat dNRB. This also applies to post-NRB plot. Would be useful to see histograms of the values to compare the distributions as well as scatterplots.

Line 221 (Figure 4.): A few questions/comment and suggestions here:

1. Why the authors chose not to present pre-NBR subplot? Given that post-NBR shows highest correlation but dNBR and RdNBR indicate weaker relationships, the question rises what is the situation with the pre-NBR estimates.

2. Do the plots show all of the validation data? To me it seems that negative values were cut off.

3. I think it would be very useful to see histograms of the estimated burn severity indices. In particular, distributions of global MOSEV values (could be data from one year, 2019) and distributions of data used for comparison with Landsat, showing both MOSEV and Landsat values. This would show if the selected comparison regions are representative of global values and any biases between MOSEV and Landsat-based

estimates.

4. In addition, rather than having pooled global comparison, I think it would be very interesting to split the analysis data into different land cover types. MODIS product product MCD12Q1 (which has the same projection and tiling system as MOSEV) for the year 2019 could used to determine land cover for the comparison burned pixels. It would be indeed interesting to see what the agreement between MOSEV and Landsat estimates is for forests versus grasslands and agricultural land. This suggestion is however optional as this would involve analysis of new datasets.

Lines 233 – 235: What are the algorithmic differences between MOSEV and other MODIS-based burned severity products? Also, a discussion of differences between MOSEV and other MODIS-based NBR studies should perhaps be included in the introduction.

---

## Referee Comment (RC2) · Anonymous Referee #2 · 4 Feb 2021

MOSEV: a global burn severity database from MODIS (2000-2020) is a very interesting paper that highlights an available dataset including burn severities caculated with MODIS imaginery which is easily accessible. The paper evaluate its potential by comparing to results obatained from landsat images and validate the great versatility as an useful ttol on large scales

After the changes and correction made with the comments and suggestions of another reviewer, my opinion is that the document is ready to be accepted

---

## Author Response (AR1)

**Response to Referee #1 (essd-2020-341)**

**General comments**

**Comment 1:** The manuscript presents a newly developed global database of MODIS-based burn severity indices for the 21st century. I believe that such readily available dataset is an important development which would enable researchers to perform a range of analyses focused on global fire dynamics and impacts. I can certainly see myself using the dataset in the future. The manuscript is generally well structured and written, however, I have a few concerns and comments which I would like the authors to address before publication. My main critique of the manuscript is twofold. Firstly, the presented comparison of the MOSEV vs Landsat burn severity estimates is very limited and offers little in terms of actual validation and uncertainty estimation. While I do realize that validating such a dataset is difficult to say the least, but with little extra work the authors could help the potential users of the MOSEV dataset to gauge magnitude of uncertainties and biases associated with the burn severity estimates. In particular, it would be useful to see distributions of the burn severity indices in addition to the presented comparison data. Secondly, the manuscript lacks discussion on what typical satellite-based pre and post- burn NBR (and hence dNBR or RdNBR) values are and how they relate to biomass consumption/fire severity as measured by field studies or any other methods.

*Response: Thank you for your positive evaluation and comments. We hope the responses to your comments and the revised version of the document meet your demands. We have addressed the two Reviewer #1 critiques as follows:*

*(1) We have modified the figure (Fig. R1) and included a new table (Table R1) to provide the distributions, mean values and % of negative and positive values of the three burn severity indices (post-burn NBR, dNBR and RdNBR) calculated from Landsat and obtained from MOSEV. This information was useful to inspect the biases associated with the burn severity estimates and the results and discussion were improved accordingly. In general, this new information was useful to show the little differences and biases between both sources of information (Landsat and MOSEV).*

*(2) We have enriched the manuscript including a new sub-section (4.2 Indices interpretation and validity) in Results and Discussion to (i) explain the biophysical meaning of the three burn severity, (ii) to indicate what fire impacts have demonstrated to correlate including $R^2$ values from previous studies carried out in different regions, and (iii) we have provided a series of dNBR and RdNBR thresholds and references that may help users to broadly interpret and classify burn severity, differentiating low, moderate and high severity classes.*

[Figure]

*Figure R1 (Fig.4 in the manuscript). Gaussian kernel densities (left) and scatterplots (right) Scatterplots showing the relationships between the burn severity data included in the MOSEV database (dNBR, RdNBR and post-burn NBR spectral indices) and the equivalent obtained from 13 Landsat-8 scenes randomly distributed across the globe (n = 32,163). R: Pearson´s correlation coefficient. Dens. = density of points. See table A1 for further information.*

*Table R1 (Table 2 in the manuscript). Mean values, and proportion of positive and negative values of the burn severity spectral indices included in the MOSEV database (dNBR, RdNBR and post-burn NBR spectral indices) and the equivalent obtained from 13 Landsat-8 scenes randomly distributed across the globe (n = 32,163).*

| Layer | MOSEV (mean) | Landsat (mean) | MOSEV (% positive) | Landsat (% positive) | MOSEV (% negative) | Landsat (% negative) |
|---|---|---|---|---|---|---|
| dNBR | 210.3 | 216.7 | 92.0 | 98.8 | 8.0 | 1.2 |
| RdNBR | 616.0 | 651.5 | 92.0 | 98.8 | 8.0 | 1.2 |
| postNBR | -68.4 | -54.7 | 27.4 | 29.4 | 72.6 | 70.6 |

**Specific comments**

**Comment 2:** Lines 55 – 57: The sentence is the only place that I could see where a quantitative estimate of how well satellite-based burn severity estimates relate to field data. This is key information and the discussion should be expanded. It is currently not clear if these published dNBR and RdNBR validations against field estimates of biomass consumption and plant mortality apply to Landsat data? Where ther any MODIS data-based validations? Also please provide estimates for different environments if available. This information is critical in supporting the undertaken comparison of MOSEV dataset with Landsat estimates (which serves as the only validation at this point).

*Response: According to this comment, and to the general comment of the Reviewer #1, we have specified in the introduction that validations against field estimates were done with Landsat data, and we have included a new sub-section in Results and Discussion (4.2 Indices interpretation and validity) to provide details on correlations of spectral indices obtained from Landsat with field measurements of biomass consumption and related impacts such as plant mortality. Although numerous studies have reported the relationships between Landsat burn severity indices and field data, we have not found direct validations of MODIS burn severity indices with field information. This could be consequence of the complexity and arduousness of quantifying severity metrics in 500 m x 500m field plots.*

**Comment 3:** Line 85 (Figure 1) What does "Fires Detected" conditional block represent in the flowchart? Is this simply checking if MCD64A1 tile contain burned pixels of something else? If additional filtering to MCD64A1 data was applied, what were the implications for MOSEV coverage in comparison to MCD64A1 burned area?

*Response: We refer to the detection of burned pixels in the MCD64A1 tile. Thus, we have modified the figure (Fig. R2) to replace "Fires detected" per "Burned area detected".*

[Figure]

*Figure R2 (Fig. 1 in the manuscript). Methodology flowchart used for building the MOSEV database (2000-present). MOD09A1 and MYD09A1 are 8-day reflectance products at 500 m from MODIS Terra and Aqua respectively. MCD64A1 is the monthly MODIS burned area product at 500 m spatial resolution. NBR, dNBR and RdNBR are burn severity spectral indices (Normalized Burn Ratio, difference of the NBR and Relativized dNBR respectively).*

**Comment 4:** Lines 100 – 105: From the description it is not clear why burned area date uncertainty is inflated by 8 days in Eq. 3, but the same is not done when determining preNBR date in Eq. 2. Could authors explain this.

*Response: We have added 8 days to the post-fire date + uncertainty because the database was built using Terra-Aqua composites, which selects the highest quality reflectance values of an 8-day period (8-day composites) (L67-69; L103 of the submitted manuscript). Thus, to ensure that values of post-fire pixels correspond to a date after the fire date + uncertainty is necessary to add 8 days. In the case of the pre-fire date we have not this problem, because composites are made with the information of previous 8-days to the fire date – uncertainty, and therefore all of them should be unburned. Figures R3 and R4 may help to clarify this.*

[Figure]

Search: >0 days

*Figure R3: Graphical representation of a hypothetical selection of pre- and post-burn MODIS composite dates without adding 8 days to the post-burn MCD64A1 dates + uncertainty. Note that in this case the MODIS sensing period (to build the composites) for the post-burn situation may overlap the MCD64A1 burn date.*

[Figure]

Search: >0 days

*Figure R4: Graphical representation of the selection of pre- and post-burn MODIS composite dates in the MOSEV database. In this case, adding 8 days to the post-burn MCD64A1 dates we ensure that values in both pre- and post-fire MODIS composites are from the pre- and post-burn situation respectively.*

**Comment 5:** Lines 163 – 170: The section presents mathematical ranges of the data, however it would be also very interesting to see distributions of actual retrieved values, perhaps in the results section. In particular, what proportion of total dNBR and RdNBR estimates have negative values and what is the interpretation of this result. This is very interesting as MCD64A1 burned area algorithm uses difference in reflectance of bands p5 and p7 (p7 is also used by MOSEV) to determine burned area in the first place.

*Response: Although not common, negative values in dNBR and RdNBR are feasible within burned areas. In view of the interest of knowing the proportion of negative and positive values, we have included this information in a new figure (Fig. R5) and table (Table R1). Moreover, we have discussed about the presence of negative values in the MOSEV product. Specifically, we stated that" the higher temporal resolution of MODIS imagery used to build the database enables to use pre-fire information very close to the burning event. This has a significant influence in the pre-burn NBR values, which typically decrease as the fire season approaches and drought conditions are more severe (Wang et al., 2008). This fact explains the lower pre-burn NBR values and the higher proportion of negative values in the MOSEV*

*product compared with Landsat (Table R2), which unavoidably lead to some differences in dNBR and RdNBR values and could contribute to a higher proportion of negative values in these indices in the MOSEV database (Table R1)"*

[Figure]

*Figure R5 (Figure A1 in the manuscript). Gaussian kernel densities (left) and scatterplots (right) showing the relationships between the pre-burn NBR spectral index included in the MOSEV database and the equivalent obtained from 13 Landsat-8 scenes randomly distributed across the globe (n = 32,163). R: Pearson´s correlation coefficient. Dens. = density of points.*

*Table R2 (Table A2 in the manuscript). Mean values, and proportion of positive and negative values of the pre-burn NBR spectral index included in the MOSEV database and the equivalent obtained from 13 Landsat-8 scenes randomly distributed across the globe (n = 32,163).*

| Layer | MOSEV (mean) | Landsat (mean) | MOSEV (% positive) | Landsat (% positive) | MOSEV (% negative) | Landsat (% negative) |
|---|---|---|---|---|---|---|
| preNBR | 128.5 | 161.0 | 72.2 | 82.7 | 27.8 | 17.3 |

**Comment 6:** Line 178: "no fires were detected" is ambiguous as the term is usually associated with active fire satellite products. Was active fire detection data used in the study? If not, perhaps "no burned area pixels" would be more clear.

*Response: We have done the change proposed by the Reviewer #1 for clarity.*

**Comment 7:** Lines 202 – 203: I'm not sure if the scatterplots (Fig. 4) make it possible to tell whether there's a positive bias in MOSEV estimates vs Landsat. To me it seems that at lower values MOSEV estimates are lower, and that this results in the slopes of linear fits seen in Fig. 4. For dNBR subplot in particular I see a dense cluster of negative MOSEV dNBR values which correspond to low (but positive) Landsat dNBR. This also applies to post-NRB plot. Would be useful to see histograms of the values to compare the distributions as well as scatterplots.

*Response: We have included the abundance distributions of the three burn severity indices in the figure (Fig. R1) and mean values in a table (Table R1), and we have modified the text in Results and Discussion accordingly.*

**Comment 8:** Why the authors chose not to present pre-NBR subplot? Given that post-NBR shows highest correlation but dNBR and RdNBR indicate weaker relationships, the question rises what is the situation with the pre-NBR estimates.

*Response: We prefer not to present the pre-NBR subplot in the main body of the manuscript because the pre-fire NBR is not a burn severity index. However, in view of this comment, and to provide more insights on the relationships between Landsat and MOSEV indices we have included the pre-burn NBR distributions, scatterplots and mean values in the Appendix (Figure R5; Table R2). The discussion of the manuscript was improved considering the results shown in Fig. R5.*

**Comment 9:** Do the plots show all of the validation data? To me it seems that negative values were cut off.

*Response: We have modified the plots to show all the validation data, even negative values (Fig. R1; Fig. R5).*

**Comment 10:** I think it would be very useful to see histograms of the estimated burn severity indices. In particular, distributions of global MOSEV values (could be data from one year, 2019) and distributions of data used for comparison with Landsat, showing both MOSEV and Landsat values. This would show if the selected comparison regions are representative of global values and any biases between MOSEV and Landsat-based estimates.

*Response: We have included in the figure (Fig. R1) panels showing the data distribution for MOSEV and Landsat spectral indices calculated with our validation dataset (n=32,163). Moreover, we have provided information of the mean values of the spectral indices (Table R1 and R2). We think our validation dataset is representative, as it includes burned areas from different regions across the globe frequently affected by fires (Table A1 in the manuscript).*

**Comment 11:** In addition, rather than having pooled global comparison, I think it would be very interesting to split the analysis data into different land cover types. MODIS product MCD12Q1 (which has the same projection and tiling system as MOSEV) for the year 2019 could used to determine land cover for the comparison burned pixels. It would be indeed interesting to see what the agreement between MOSEV and Landsat estimates is for forests versus grasslands and agricultural land. This suggestion is however optional as this would involve analysis of new datasets.

*Response: We appreciate the reviewer suggestion and we think this is an interesting and laborious work that can be addressed in future studies. Nevertheless, until more extensive validations are done, our dataset of 13 Landsat scenes with 32,163 burned pixels at 500 m × 500 m spatial resolution, provides highly valuable information on the global relationships between Landsat and MOSEV data.*

**Comment 12:** Lines 233 – 235: What are the algorithmic differences between MOSEV and other MODIS-based burned severity products? Also, a discussion of differences between MOSEV and other MODIS-based NBR studies should perhaps be included in the introduction.

*Response: The algorithms used to calculate the burn severity indices from the MOSEV database are shown in Equations. 2 to 5. These are standard burn severity indices, well known in the field of fire ecology. Other MODIS-based NBR studies were cited in the introduction (e.g. Veraverbeke et al., 2011; Rahman et al., 2018), but we preferred not provide the details of each work for conciseness.*

**Response to Referee #2 (essd-2020-341)**

**General comment**

**Comment 1:** MOSEV: a global burn severity database from MODIS (2000-2020) is a very interesting paper that highlights an available dataset including burn severities calculated with MODIS imagery which is easily accessible. The paper evaluates its potential by comparing to results obtained from Landsat images and validate the great versatility as a useful tool on large scales. After the changes and correction made with the comments and suggestions of another reviewer, my opinion is that the document is ready to be accepted.

*Response: Thank you for your positive evaluation and comment. We have addressed all the comments of the other Reviewer and we hope the revised version of the document meet your demands.*

**References**

Rahman, S., Chang, H., Hehir, W., Magilli, C., Tomkins, K.: Inter-Comparison of Fire Severity Indices from Moderate (Modis) and Moderate-To-High Spatial Resolution (Landsat 8 & Sentinel-2A) Satellite Sensors, IGARSS 2018 - 2018 IEEE International Geoscience and Remote Sensing Symposium, Valencia, Spain, https://doi.org/10.1109/IGARSS.2018.8518449, 2018.

Veraverbeke, S., Lhermitte, S., Verstraeten, W. W., Goosens, R.: A time-integrated MODIS burn severity assessment using the multi-temporal differenced normalized burn ratio (dNBRMT), Int. J. Appl. Earth Obs. Geoinformation, 13, 52-58. https://doi.org/10.1016/j.jag.2010.06.006, 2011.

Wang, L., Qu, J.J., Hao, X: Forest fire detection using the normalized multi-band drought index (NMDI) with satellite measurements, Agric. For. Meteorol., 148, 1767-1776. https://doi.org/10.1016/j.agrformet.2008.06.005, 2008.